# *Schizothorax prenanti* Heat Shock Protein 27 Gene: Cloning, Expression, and Comparison with Other Heat Shock Protein Genes after Poly (I:C) Induction

**DOI:** 10.3390/ani12162034

**Published:** 2022-08-10

**Authors:** Jianlu Zhang, Kunyang Zhang, Jiqin Huang, Wei Jiang, Hongying Ma, Jie Deng, Hongxing Zhang, Wanchun Li, Qijun Wang

**Affiliations:** 1Shaanxi Key Laboratory of Qinling Ecological Security, Shaanxi Institute of Zoology, Xi’an 710032, China; 2Key Laboratory for Hydrobiology in Liaoning Province, College of Fisheries and Life Science, Dalian Ocean University, Dalian 116023, China; 3Hanzhong Nanzheng District Aquatic Product Workstation, Hanzhong 723100, China

**Keywords:** antiviral immunity, gene expression, heat shock protein 27, polyinosinic-polycytidylic acid, *Schizothorax prenanti*

## Abstract

**Simple Summary:**

*Schizothorax prenanti* is a valuable cold-water fish that is commercially farmed in southwest China. Numerous aquaculture farmers have recently adopted high-density farming to achieve greater economic benefits, but this has rendered *S*. *prenanti* more susceptible to microbial pathogens and resulted in economic losses. Hence, the immune mechanisms of *S*. *prenanti* against pathogens should be investigated. Heat shock proteins (Hsps) comprise a family of molecular chaperones that are involved in immune pathways. Here, we identified and cloned the cDNA encoding S*pHsp27* gene and detected its tissue distribution and using polyinosinic-polycytidylic acid [Poly (I:C)] as a viral analog to challenge the fish. We also explored the expression of *SpHsp27*, *SpHsp60*, *SpHsp70*, and *SpHsp90* in four immune organs from fish that were injected with Poly (I:C). We found that Poly (I:C) induced *SpHsp27* expression in all of these tissues, and significantly up-regulated most *SpHsps* genes compared with controls that were injected with phosphate-buffered saline. However, temporal expression and tissues were organ-specific. The present findings will help to further clarify the roles of Hsp genes in the mechanisms of antiviral immunity in fish.

**Abstract:**

We identified and cloned cDNA encoding the heat shock protein (Hsp) 27 gene from *Schizothorax prenanti* (*SpHsp27*), and compared its expression with that of *SpHsp60*, *SpHsp70*, and *SpHsp90* in the liver, head kidney, hindgut, and spleen of *S. prenanti* that were injected with polyinosinic-polycytidylic acid [Poly (I:C)]. The *SpHsp27* partial cDNA (sequence length, 653 bp; estimated molecular mass, 5.31 kDa; theoretical isoelectric point, 5.09) contained an open reading frame of 636 bp and a gene encoding 211 amino acids. The SpHsp27 amino acid sequence shared 61.0–92.89% identity with Hsp27 sequences from other vertebrates and *SpHsp27* was expressed in seven *S. prenanti* tissues. Poly (I:C) significantly upregulated most *SpHsps* genes in the tissues at 12 or 24 h (*p* < 0.05) compared with control fish that were injected with phosphate-buffered saline. However, the intensity of responses of the four *SpHsps* was organ-specifically increased. The expression of *SpHsp27* was increased 163-fold in the head kidney and 26.6-fold *SpHsp27* in the liver at 24 h after Poly (I:C) injection. In contrast, *SpHsp**60* was increased 0.97–1.46-fold in four tissues and *SpHsp**90* was increased 1.21- and 1.16-fold in the liver and spleen at 12 h after Poly (I:C) injection. Our findings indicated that Poly (I:C) induced S*pHsp27*, *SpHsp60*, *SpHsp70*, and *SpHsp90* expression and these organ-specific *SpHsps* are potentially involved in *S. prenanti* antiviral immunity or mediate pathological process.

## 1. Introduction

Heat shock proteins (Hsps) are a class of highly conserved proteins (molecular mass, 16–100 kDa) that are produced in prokaryote and eukaryote cells generally in response to stress [1]. Under normal conditions, Hsps help to ensure correct protein folding during synthesis and repair misassembled proteins [2]. The gene expression of *Hsps* is usually induced by elevated temperatures. However, many Hsps play pivotal roles in intracellular anti-stress and immune processes and are associated with autoimmune diseases [3,4]. They have been widely applied as biomarkers of biological, chemical, and physical stress [5,6]. Heat shock proteins are molecular chaperones that are categorized into Hsp100, Hsp90, Hsp70, Hsp60, and Hsp27 subfamilies of small heat shock proteins, (sHsps) according to their protein sequence homology and molecular mass. Small Hsps belong to a highly conserved class of Hsps consisting of 16–42 kDa proteins [7,8]. The vital functions of sHsps include the degradation of misfolded proteins [9] and the modulation of cell growth, apoptosis, as well as cytoskeleton elements [10].

Various organisms express evolutionarily conserved Hsps that protect against various environmental stressors [3,4]. Heat shock proteins have notable constitutive functions that are essential for protein metabolism in unstressed cells [11,12]. Therefore, heat shock protein genes (*Sp**Hsps*) distribution in different tissues of *Schizothorax prenanti* should be investigated under normal conditions.

Fish Hsps have been assessed in coho salmon (*Oncorhynchus kisutch*) [13], gilthead sea bream (*O**. mykiss*), rainbow trout (*Sparus aurata*) [14,15], common carp (*Cyprinus carpio*) [16], miiuy croaker (*Miichthys miiuy*) [17], and grass carp (*Ctenopharyngodon idella*) [18,19]. Li et al. (2015) cloned *Sp*Hsc70 and *Sp*Hsp70 and detected changes in their tissue-specific expression in response to challenge with *Aeromonas hydrophila* [20]. Pu et al. (2016) cloned and detected changes in *SpHSP90* expression after *Streptococcus agalactiae* infection [21]. However, *SpHSP27* has not yet been identified in fish.

Polyinosinic-polycytidylic acid [Poly (I:C)] is a viral analog that stimulates the immune system and induces a characteristic inflammatory response in organisms [22] and it has been used to study the antiviral mechanism of aquatic animals [23,24,25,26,27,28,29,30,31,32,33,34,35,36]. The role of Poly (I:C) in enhancing the natural immunity and reducing disease risk has attracted attention. Moreover, the immune responses to Poly (I:C) differs among fish species [23,24,25,26]. However, its effects on different immune tissues of *S. prenanti* have remained unknown. Heat shock protein 27 has been identified in zebrafish (*Danio rerio*), desert fish (*Poeciliopsis lucida*), and channel catfish (*I**ctalurus punctatus*) [27,28,29]. However, the function of *Hsp27* in these fish in response to viral challenge remains unclear.

*Schizothorax prenanti* belongs to the Schizothoracinae subfamily of the Cyprinidae family. It is mainly distributed in the upper reaches of the Yangtze River, including the Western Plateau of China. This economically important cold-water fish has been artificially cultivated and marketed for consumption at approximately USD6.7 per kilogram. Known as Ya-fish and Yang-fish in Sichuan and Shaanxi Provinces, respectively, this fish has been listed as a protected species in these regions due to the continual decline of wild populations [21,30,31].

In general, the liver, head kidney, and hindgut of the fish are regarded as immune organs that are central to immune responses [32,33] and Hsp gene expression has been assessed in fish spleens. Understanding how *SpHsp* expression varies after induction by Poly (I:C) might help to clarify the anti-viral and immune mechanisms of *S. prenanti* and a scientific basis for an in-depth study of *SpHsps*. Increasing numbers of aquaculture farmers have adopted high-density farming to achieve greater economic benefits, but this strategy renders *S*. *prenanti* more susceptible to microbial pathogens such as *Streptococcus agalactiae* [21] and *Aeromonas hydrophila* [20,31]. Although viral diseases of *S*. *prenanti* remain unknown, the immune response of this fish to viruses should be determined in advance. Therefore, we identified and cloned cDNA encoding *SpHsp27* for the first time and analyzed *SpHsp27*, *60*, *70*, and *90* responses to Poly (I:C) by quantitative real-time RT-PCR (qRT-PCR). Our findings will help to further clarify the roles of Hsp genes that are involved in the antiviral immunity mechanisms of fish.

## 2. Materials and Methods

### 2.1. Animal Treatment

We obtained healthy cultured *S. prenanti* (121.7 ± 28 g) in May 2020 from Qunfu Yang-fish professional breeding cooperative, a commercial breeding aqua-farm in Hanzhong city, Shaanxi, China. The fish were maintained in glass tanks with a size of 60 × 30 × 40 cm^3^, and with aerated tap water at a temperature of 20 ± 1 °C. The tank filter was cleaned and about a quarter of the aerated tap water was replaced daily. The fish were fed with a commercial feed (floating compound feed with crude protein ≥32%, crude fat ≥3%) at a rate of 2% of their body weight twice daily. After 10 days acclimation, the fish were challenged with an intraperitoneal injection of Poly (I:C) (5 mg/kg body weight, P1530, sigma, St Louis, MO, USA), and the control group was injected with PBS at the same amount. To understand the expression of *SpHsps* in response to Poly (I:C) challenge, anatomical samples of Poly (I:C)-stimulated group and 5 mg/kg body weight PBS-stimulated group were taken at 12 h and 24 h after stimulation. The fish were anesthetized with eugenol at a concentration of 80 mg/L for 3 min before being dissected. The heart, liver, head kidney, hindgut, muscle, intraperitoneal fat, and spleen were collected from four healthy *S.prenanti* for examining the tissue distribution of *SpHsp27*. The liver, head kidney, hindgut, and spleen were collected from four *S.prenanti* of the PBS–injection and Poly (I:C) treatment at 12 and 24 h, respectively. The dissected tissues were preserved in liquid nitrogen for RNA extraction.

### 2.2. RNA Extraction and cDNA Synthesis

The tissue total RNA was extracted using the Trizol (Invitrogen, Carlsbad, USA) method. The total RNA concentration and purity were determined by RNA electrophoresis and the optical density absorption ratio (A260/280) in the Nanodrop One spectrophotometer (Thermo Fisher Scientific, Wilmington, NC, USA). First-stand cDNA was synthesized using the RevertAid First Strand cDNA Synthesis Kit (Thermo Scientific, Vilnius, Lithuania), according to the method that was recommended by the manufacturer.

### 2.3. Partial cDNA Cloning of the SpHsp27

According to the transcriptome sequencing of *S. prenanti*, the specific *Hsp27* primers were designed (Table 1, Hsp27-F and Hsp27-R) and synthetized by Tsingke Biotechnology Co., Ltd. (Xi’an, China). Liver cDNA was used as a template for *SpHsp27* amplification, with Primerstar^®^ Max DNA polymerase (TaKaRa, Dalian, China). The PCR program was as follows: 35 cycles at 98 °C for 10 s, 55 °C for 15 s, and 72 °C for 40 s. The PCR products were tailed A with DNA A-Tailing Kit (TaKaRa, Dalian, China). The obtained PCR products were ligated into a pMD19-T vector (TaKaRa, Dalian, China), and transformed into competent *Escherichia coli* DH5α cells. The positive bacteria clones were sequenced.

### 2.4. Sequence Analysis

The open reading frame (ORF) of *S. prenanti Hsp27* (*SpHsp27*) was identified using the ORF Finder (http://www.ncbi.nlm.nih.gov/gorf/gorf.html (accessed on 10 May 2022)). The isoelectric point and molecular weight were predicted (https://web.expasy.org/compute_pi/, (accessed on 11 May 2022)). The secondary structure composition of *Sp*Hsp27 protein was predicted (https://www.novopro.cn/tools/secondary-structure-prediction.html (accessed on 11 May 2022)). The three-dimensional structure of *Sp*Hsp27 protein was predicted (https://swissmodel.expasy.org/ (accessed on 11 May 2022)). Multiple sequence alignments were performed using the Clustal X2 [34]. A phylogenetic tree of different vertebrates Hsp27 that was based on amino acid sequences was constructed by the neighbor-joining method using MEGA 4.0 software [35].

### 2.5. Tissue Distribution of SpHsp27 mRNA under Normal Conditions

The measured tissues in the control group included the heart, liver, spleen, head kidney, muscle, intraperitoneal fat, and hindgut. Total RNA was extracted and cDNA was synthesized as described previously (“RNA extraction and cDNA synthesis”). The qRT-PCR analysis was performed using the applied Biosystems Step One Plus (Life Technologies, Foster City, USA). The gene-specific primers are listed in Table 1. *S. prenanti*-specific β-actin primers were used to normalize the cDNA quantity for each tissue sample. Quantification of *SpHsp27* and β-actin was performed in triplicate on all the samples using FastStart Essential DNA Green Master (Roche), according to the manufacturer’s instructions. The qRT-PCR data were calculated according to the 2^−ΔΔCT^ method [36].

### 2.6. Detection of the Expression Patterns Induced by Poly (I:C)

For the Poly (I:C) challenge, 8 fish were injected intraperitoneally from behind the base of the pectoral fin with Poly (I:C) (5 mg/kg). Another 4 fish were injected intraperitoneally PBS at the same volume, as a control. A total of 4 fish samples were taken at 12 h and 24 h after Poly (I:C) injection, respectively. The head kidney, liver, hindgut, and spleen were collected from each fish and used to isolate the total RNA. To detect the expression of *SpHsp27*, *SpHsp60*, *SpHsp70,* and *SpHsp90* changes after Poly (I:C) challenge, total RNA extraction, cDNA synthesis, and qRT-PCR were performed, as described previously (“Tissue distribution of *SpHsp27* mRNA in unstressed conditions”). *S. prenanti*-specific β-actin primers were used to normalize the cDNA quantity for each tissue sample.

### 2.7. Statistical Analysis

SPSS 22.0 and Graphpad Prism 5.0 software were used for data analysis and histogram, respectively. The mRNA expression levels were analyzed by using the one-way ANOVA method. All the data are presented as the mean ± standard error (*n* = 4), and the statistically significant differences between PBS control and Poly (I:C) treatment groups at each time point are expressed with asterisks: * *p* < 0.05, ** *p* < 0.01, *** *p* < 0.001 vs. corresponding control group at the time points.

## 3. Results

### 3.1. Identification, Structural, and Phylogenetic Analysis of SpHsp27

The 653-bp sequence of *SpHsp27* cDNA included an ORF of 636 bp (Figure 1). The ORF encoded a predicted protein of 211 amino acids with a calculated molecular mass of 5.31 kDa and a theoretical isoelectric point of 5.09. Figure 2a shows the distribution of strands, helixes, and coils, and Figure 2b shows the predicted 3D structure of *Sp*Hsp27 protein. The *Sp*Hsp27 amino acid sequence was the most similar to those of fish among animals and was the closest to goldfish (*Carassius auratus*) Hsp27 with 92.89% identity. The multiple sequence alignment in Figure 3 shows that *Sp*Hsp27 protein is moderately conserved and contains a crystallin domain (dark green, with light blue region) and two putative actin interacting domains (light blue).

We analyzed the phylogeny of the *Sp*Hsp27 amino acid sequences to determine the evolutionary relationships between *Sp*Hsp27 and Hsp27 from other vertebrates, based on sequences in the GenBank database (Figure 4). The results revealed high amino acid sequence identity between *Sp*Hsp27 and the Hsp27 of *Cyprinids*, particularly goldfish.

### 3.2. Tissue Distribution of SpHsp27 Expression in S. Prenanti

We quantified *SpHsp27* mRNA expression in the heart, liver, spleen, head kidney, muscle, intraperitoneal fat, and hindgut tissues from four fish using qRT-PCR to determine *SpHsp27* transcripts expression. The loading control for normalization was β-actin. Figure 5 shows the ubiquitous, but variable expression of *SpHsp27* transcripts in all seven tissues. The expression of *SpHsp27* was significantly higher in the heart and muscle than in all the other tissues, and in all other tissues except the heart, respectively (both *p* < 0.05) The expression *SpHsp27* did not significantly differ among the liver, spleen, head kidney, intraperitoneal fat, and hindgut tissues.

### 3.3. Expression of SpHsps after Challenge with Poly (I:C)

We quantified the transcripts in the liver, head kidney, hindgut, and spleen of *S. prenanti* by qRT-PCR at 12 and 24 h after injecting Poly (I:C) to determine changes in *SpHsp27*, *SpH**sp60, SpHsp70*, and *SpHsp90* expression.

#### 3.3.1. Expression of SpHsp27 after Injection of Poly (I:C)

The levels of *SpHsp27* transcripts were significantly higher in the liver and spleen tissues than those of the controls at 12 h after injecting *S. prenanti* with Poly (I:C) (*p* < 0.05, and *p* < 0.01, respectively (Figure 6) but did not significantly differ in the head kidney or the hindgut. However, *SpHsp27* was overexpressed and significantly higher in the head kidney and liver that wereinjected after Poly (I:C) injection than in the controls at 24 and 12 h (*p* < 0.001). Moreover, *SpHsp27* transcripts were significantly more abundant in the hindgut and spleen at 24 h after Poly (I:C) injection compared with that at 12 h, and PBS (*p* < 0.01).

#### 3.3.2. Expression of SpHsp60 after Challenge with Poly (I:C)

The expression of the *SpHsp60* gene did not significantly change in the four tissues at 12 h after Poly (I:C) injection compared with the controls. However, the mRNA levels in the liver and head kidney were significantly higher (*p* < 0.001) than that in the PBS controls and 12 h point after Poly (I:C) injection, whereas the those in the hindgut and spleen remained unchanged at 24 h after injection (Figure 7).

#### 3.3.3. Expression Levels of SpHsp70 after Challenge with Poly (I:C)

Figure 8 shows that *SpHsp70* transcripts were upregulated in all four tissues at 12 and 24 h after Poly (I:C) injection. The levels were significantly upregulated and downregulated in the head kidney at 12 and 24 h after Poly (I:C) injection (*p* < 0.001). The expression of the *SpHsp70* gene significantly increased in the liver and hindgut (*p* < 0.001), as well as the spleen (*p* < 0.01) at 24 h after Poly (I:C) injection, and might not have reached a peak.

#### 3.3.4. Expression of SpHsp90 after Challenge with Poly (I:C)

Figure 9 shows that *SpHsp90* mRNA was significantly upregulated at 12 h after Poly (I:C) injection (*p* < 0.001) in the head kidney and hindgut, but not significantly altered in the liver and spleen. At 24 h after Poly (I:C) injection, *SpHsp90* was significantly overexpressed in the liver (*p* < 0.001), and significantly higher in the head kidney than in the PBS control (*p* < 0.001) but did not differ from the level at 12 h after injection. These findings indicated that the increased expression of the *SpHsp90* gene persisted in *S. prenanti* that were injected with Poly (I:C). Moreover, the expression in the hindgut significantly decreased to control levels at 24, compared with 12 h after Poly (I:C) injection (*p* < 0.001). Although the spleen is a major immune organ, *SpHsp90* expression did not significantly change within 24 h of Poly (I:C) injection. We speculate that *SpHsp90* gene expression in the spleen is involved later during the immune response. However, this speculation requires further investigation.

## 4. Discussion

Small Hsps play important roles in cell homeostasis and in the immune responses of fish to a diverse range of environmental pathogens, including viruses [33,37,38,39,40]. Heat shock protein 27 is a vital member of the sHsp family and its expression is upregulated in cells that are exposed to viruses, bacteria, heat, and other stresses [41,42]. Moreover, Hsp27 participates in various signaling pathways and might be a significant therapeutic target [43]. Here, we identified and cloned *Sp**Hsp27* for the first time. Multiple sequence alignment showed that *Sp*Hsp27 protein is moderately conserved, and that its amino acid sequence has high identity with those of *Cyprinids*, particularly goldfish. The results of *Sp*Hsp27 amino acid sequencing and phylogenetic tree analysis revealed that *S. prenanti* is more closely related to goldfish. This finding was consistent with our previous findings of *Sp**Hsp6*0 [44], whereas *Sp*Hsp70 [20], Hsc70 [20], and *Sp*Hsp90 [21] were more closely related to common carp, allogynogenetic silver crucian carp (*C. auratus gibelio*), and zebrafish, respectively. Under non-stressed conditions, the expression of *SpHsp27* was maximal in the heart, followed by muscle, consistent with the studies of zebrafish [45], rainbow trout [46], and sea perch (*Lateolabrax japonicus*) [47]. Le et al. (2017) speculated that the high expression level of *Hsp27* in the muscle and heart of sea perch might be related to its important role in muscle maintenance [47]. Additionally, Robinson et al. (2010) have confirmed that *Hsp27* plays a key role in stabilizing the actin cytoskeleton within the smooth muscle and endothelial cells [48] To date, reports describing *Hsp27* in fish are scant in the literature. We found that Poly (I:C) significantly upregulated *SpHsp27* gene expression in various tissues, especially in the liver and head kidney. This differs from the findings of Stenberg et al. (2019), who found significantly downregulated *Hsp27* gene expression in the head kidney leukocytes of Atlantic salmon (*Salmo salar*) that were incubated with Poly (I:C), compared with the controls [49].

Highly-conserved, multifunctional immunogenic Hsp60 is a mitochondrial matrix protein that is found in organisms ranging from primitive prokaryotes to higher eukaryotes [50,51]. The involvement of Hsp60 in regulating immune responses and modulating signaling pathways has been characterized [52,53]. The *Hsp60* gene from pleopoda of the shrimp *Litopenaeus vannamei*, is significantly upregulated 24 h after infection with white spot syndrome virus [54]. The present findings of *SpHsp60* expression in the liver and head kidney 24 h after Poly (I:C) injection is consistent with these results. We showed here that *SpHsp60* gene expression notably remained unchanged in the hindgut and spleen within 24 h after Poly (I:C) injection. Huang et al. [54] suggested that *Hsp60* plays an important role in shrimp health, especially in promoting inflammation, as well as specific and non-specific and immune responses against bacteria and viruses. Therefore, we inferred that *Hsp60* plays a similar but organ-specific role in the anti-pathogen immunity of *S. prenanti.* However, *Hsp60* and *Hsp90* expression remains unchanged throughout progressive vibriosis, whereas the expression of *Hsp*70 and *Hsc*70 decreases in the kidney and liver tissues of sea bream (*Sparus sarba*) [55]. Eder et al. [56] showed that the response intensity of Chinook salmon (*Oncorhynchus tshawytscha*) *Hsp60* depends on insecticide induction and is organ-specific. The present findings suggest that the intensity of the *SpHsp60* gene response is also organ-specific, and that *SpHsp60* expression in the liver and head kidney might be an early marker of viral infection in *S. prenanti*.

The Hsp70 family is encoded by non-inducible heat shock cognate protein 70 kDa (*Hsc70*) and inducible Hsp 70 kDa (*Hsp70*) genes [57]. Various types of stress induce *Hsp70* expression, whereas *Hsc70* expression remains unchanged or slightly upregulated [57,58]. Therefore, we investigated the *SpHsp70* gene. Heat shock protein 70 suppresses apoptosis by directly associating with Apaf-1 [59]. The yellow drum (*Nibea albiflora*) *Hsp70* gene (*NaHsp70*) is significantly upregulated in the liver at 12 h after Poly (I:C) infection, and peaks at 24 h [60]; our results were similar to these. However, *NaHsp70* mRNA expression in the *S. prenanti* spleen was significantly upregulated at 12, 24, and 36 h, whereas *SpHsp60* was not expressed at 12 h, but significantly upregulated at 24 h after Poly (I:C) injection. Moreover, *NaHsp90* expression peaks at 12 h in the liver, whereas *SpHsp90* expression in the liver peaked at 24 h after Poly (I:C) injection. Temporal expression also differed between *NaHsp90* and *SpHsp90* in spleen tissues. We speculate that this could be due to species-specific differences. The *Hsp70* and *Hsp90* genes in the two fish species were activated by Poly (I:C), although the timing of significant upregulation differed. *Streptococcus agalactiae* significantly induces *SpHsp90* gene expression in the blood, liver, spleen, and trunk kidney, and peaks at 24 h in the liver, spleen, and kidney of *S. prenanti* [21]. Our results somewhat differed, suggesting that *SpHsp90* gene expression is pathogen-specific. In summary, the *SpHsp90* gene might be involved in resistance to bacterial and viral challenge and might be an early marker of infection. Although the spleen is a main immune organ, we found that *SpHsp90* expression did not significantly change within 24 h of Poly (I:C) injection. We speculated that *SpHsp90* expression in the spleen is involved in the immune response at a later time. However, this awaits confirmation by further study.

The expression of *SpHsp27* at 24 h after Poly (I:C) injection was higher in the head kidney than in other tissues, suggesting that it is most suitable for detecting *SpHsp27* 24 h after viral infection. The expression profiles of the other three *SpHsp* genes were similar. For example, the hindgut is suitable for assessing *SpHsp60* and *SpHsp70* and the liver is suitable for assessing *SpHsp60* and *SpHsp90* at 24 h after viral injection. We also found that the liver and spleen were the most sensitive to Poly (I:C) at 12 h after injection and were thus suitable for evaluating *SpHsp27* expression, whereas *SpHsp70* could be evaluated in the head kidney, and *SpHsp90* could be evaluated in the head kidney and hindgut. Furthermore, regulation of the expression of the four *SpHsp* genes was quite complex. Many immune-related genes and signaling pathways are expressed at the transcriptional and translational levels.

Our findings showed that *S. prenanti* is very sensitive to Poly (I:C). The significantly increased expression of *SpHsp27*, *SpHsp60*, *SpHsp70*, and *SpHsp90* after Poly (I:C) injection indicated that they potentially play important roles as molecular chaperones under virus-induced stress. In summary, Poly (I:C) induced the expression of four *SpHsps* with different tissue-specific and temporal profiles, indicating coordinated action against Poly (I:C). Further studies are needed to determine the role of *SpHsps* in the innate immune defense system against viral pathogens, or mediate pathological process, and in the involved signaling pathways.

## 5. Conclusions

In the present study, the partial cDNA encoding the full ORF of *SpHsp27* genes of *S. prenanti* was successfully cloned and characterized. Phylogenetic analysis showed that the *Sp*Hsp27 protein is most closely related to Hsp27 from goldfish. Multiple sequence alignment showed that the *Sp*Hsp27 is moderately conserved. The *Sp**Hsp27* gene expressed in all tissues that were examined herein was cloned for the first time. Poly (I:C) organ-specifically-induced S*pHsp27*, *SpHsp60*, *SpHsp70*, and *SpHsp90* that are potentially involved in antiviral immunity or mediate pathological process. Notably, S*pHsp27* was the most sensitive to Poly (I:C) followed by S*pHsp70.*

## Figures and Tables

**Figure 1 animals-12-02034-f001:**
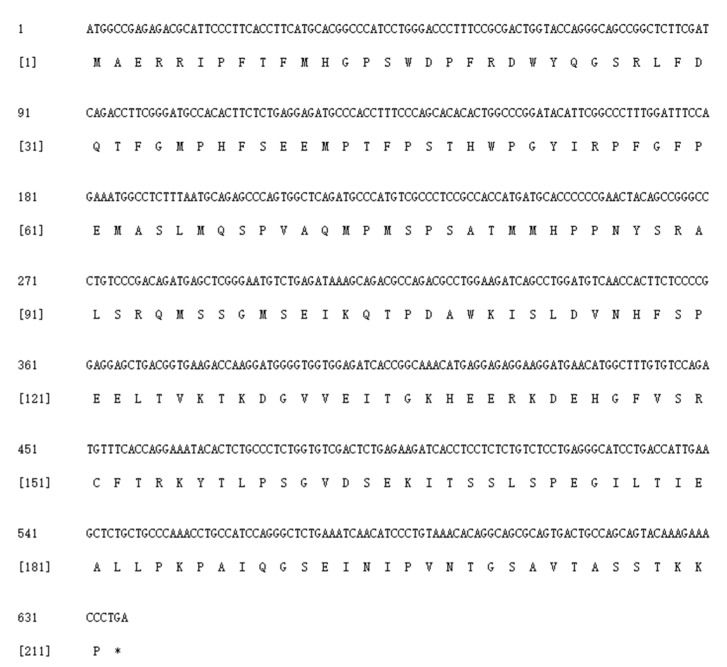
Nucleotide and deduced amino acid sequences of *Sp*Hsp27. * Stop codon.

**Figure 2 animals-12-02034-f002:**
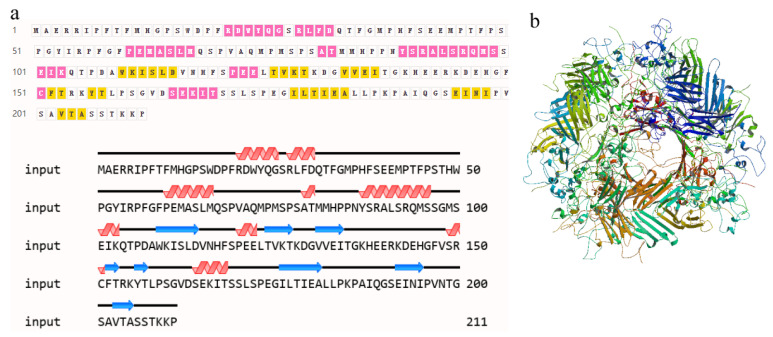
Predicted secondary structure and 3D-structural models of *Sp*Hsp27 protein. (**a**) Predicted secondary structure of *Sp*Hsp27 protein. Helix, ■; coil, ☐; strand, ■. (**b**) Three-dimensional model structures of predicted *Sp*Hsp27 protein.

**Figure 3 animals-12-02034-f003:**
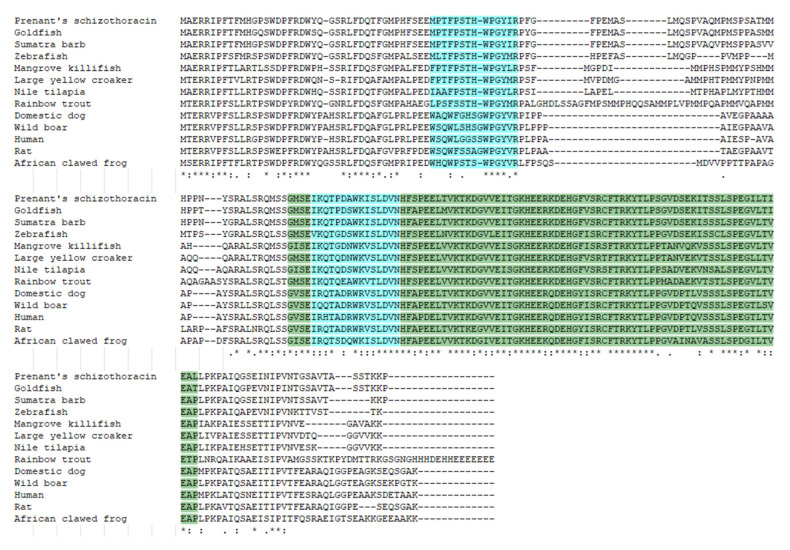
Alignment of amino acid sequences of Prenant’s schizothoracin (*Schizothorax prenanti*) Hsp27 with Hsp27 proteins from goldfish (GenBank accession no. ABI26639), Sumatra Barb (GenBank accession no. XP_043096630), zebrafish (GenBank accession no. NP_001008615), Mangrove killifish (GenBank accession no. AEM65174), Large yellow croaker (GenBank accession no. ADX98507), Nile tilapia (GenBank accession no. AFY13335), Rainbow trout (GenBank accession no. BAF80897), Domestic dog (GenBank accession no. AXQ88113), Wild boar (GenBank accession no. AAV54182), Human (GenBank accession no. BAB17232), Rat (GenBank accession no. AAA41353), and African clawed frog (GenBank accession no. ABF17872). Asterisk (⁎), identical; colon (:), conserved; and dot (.), semi-conserved residues. The conserved crystallin domain is shown in dark green (including light blue region). There are two putative actin interacting domains that are shown in light blue.

**Figure 4 animals-12-02034-f004:**
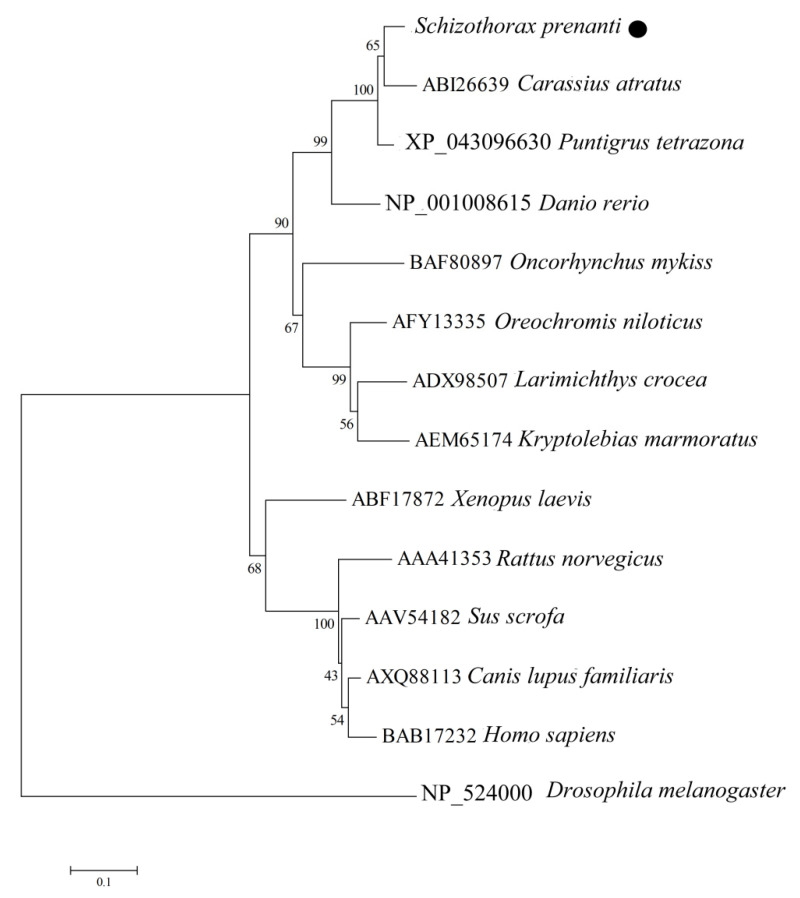
Phylogenetic tree of relationships between *S. prenanti* Hsp27 and other vertebrates. The tree was constructed by the neighbor-joining method using MEGA 4.0 software. The numbers at the nodes indicate proportions of bootstrapping after 1,000 replications. ● *Schizothorax prenanti* Hsp27.

**Figure 5 animals-12-02034-f005:**
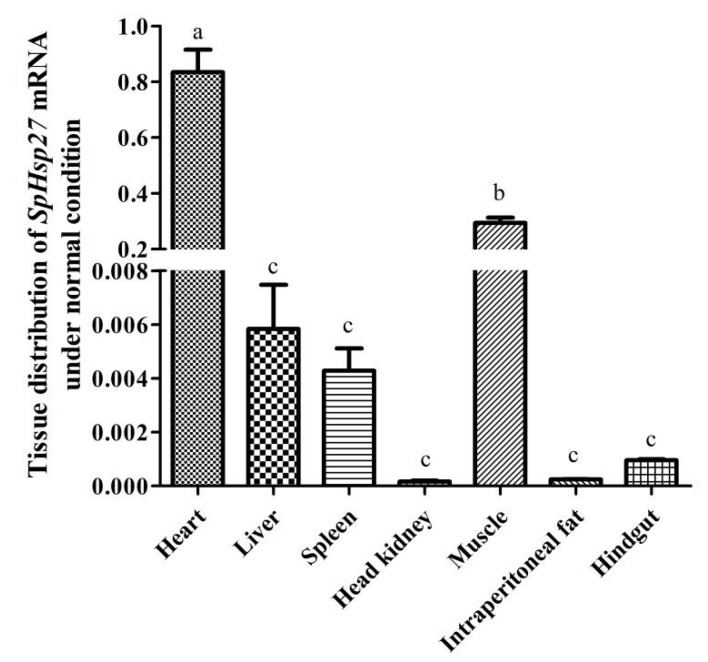
Abundance of *SpHsp27* transcripts in the heart, liver, spleen, head kidney, muscle, intraperitoneal fat, and intestine of *S. prenanti* as determined by qRT-PCR. The loading control for normalization was β-actin. (a, b, c) Means with different letters are significantly different from each other (*p* < 0.05). Values are shown as the means ± standard error (*n* = 4). Error bars, standard error of the means (*n* = 4 fish per group).

**Figure 6 animals-12-02034-f006:**
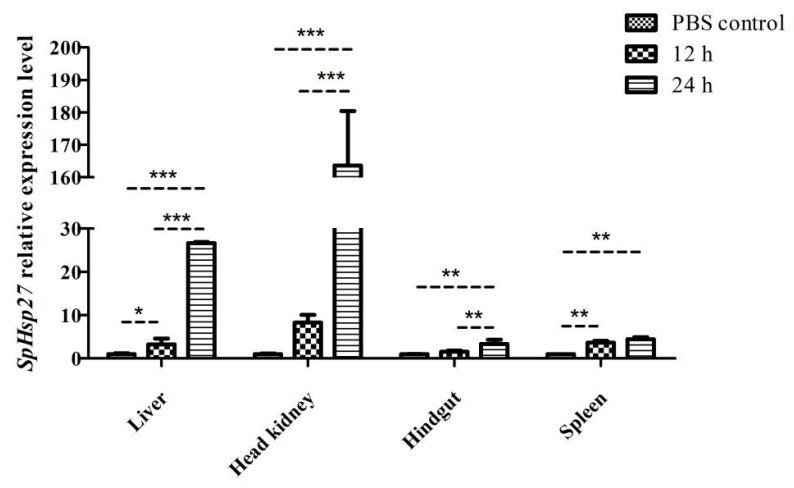
Levels of *SpHsp27* transcripts that were determined by qRT-PCR in the liver, head kidney, hindgut, and spleen of *S. prenanti* at 12 and 24 h after Poly (I:C) injection. Values were normalized using β-actin. Statistically significant differences between the groups are indicated by asterisks (* *p* < 0.05, ** *p* < 0.01, *** *p* < 0.001. PBS, phosphate-buffered saline (*n* = 4 fish per group).

**Figure 7 animals-12-02034-f007:**
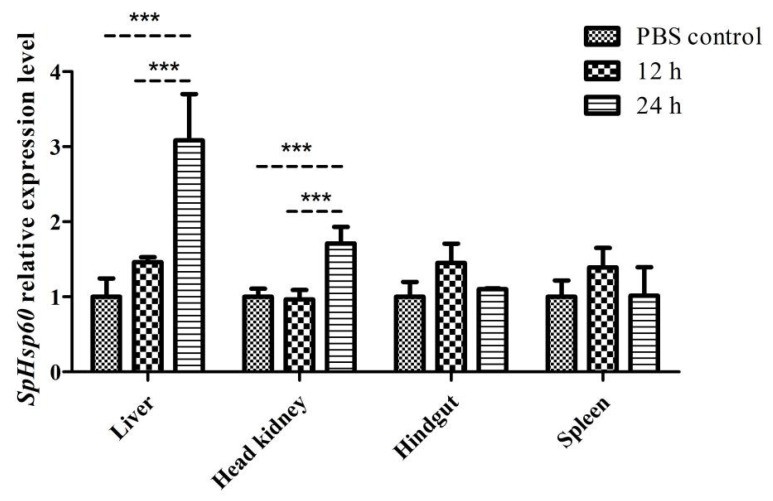
The levels of *SpHsp27* transcripts in the liver, head kidney, hindgut, and spleen of *S. prenanti* at 12 and 24 h after Poly (I:C) injection that was determined by qRT-PCR. The values were normalized using β-actin. The expression of β-actin was used as the normalization for qRT-PCR. Statistically significant differences between the groups are indicated by asterisks (*** *p* < 0.001. PBS, phosphate-buffered saline (*n* = 4 fish per group).

**Figure 8 animals-12-02034-f008:**
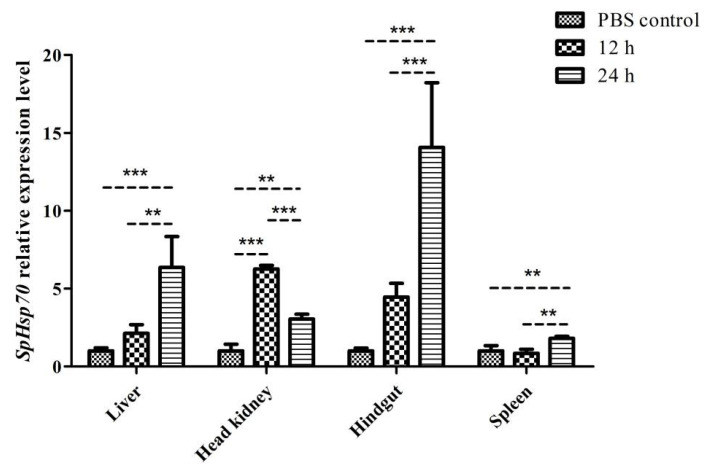
The levels of *SpHsp27* transcripts in the liver, head kidney, hindgut, and spleen of *S. prenanti* at 12 and 24 h after Poly (I:C) injection as determined by qRT-PCR. The values were normalized using β-actin. The expression of β-actin was used as the normalization for qRT-PCR. Statistically significant differences between the groups are indicated by asterisks (** *p* < 0.01, *** *p* < 0.001). PBS, phosphate-buffered saline (*n* = 4 fish per group).

**Figure 9 animals-12-02034-f009:**
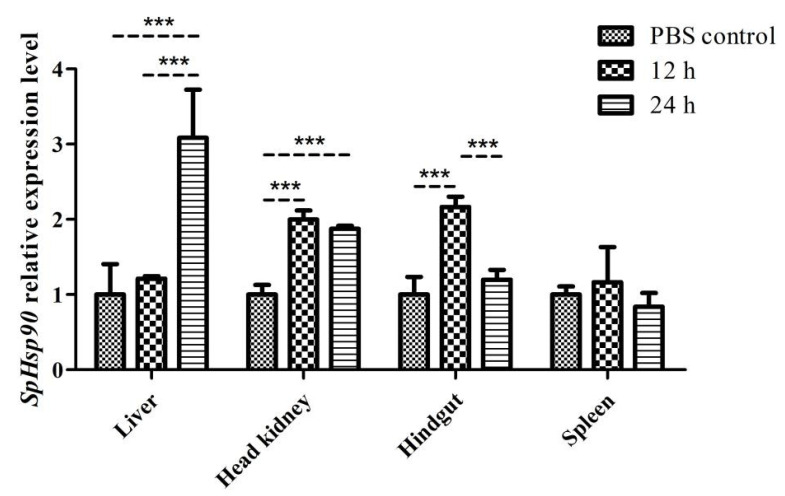
Levels of *SpHsp27* transcripts in the liver, head kidney, hindgut, and spleen of *S. prenanti* at 12 and 24 h after Poly (I:C) injection that was determined by qRT-PCR. The values were normalized using β-actin. Statistically significant differences between the groups are indicated by asterisks (*** *p* < 0.001). PBS, phosphate-buffered saline (*n* = 4 fish per group).

**Table 1 animals-12-02034-t001:** The primers used for *SpHsp27* clone and qRT-PCR.

Primer	Sequence (5′-3′)	Annealing Temperature (°C)	Size (bp)
Primers for cloning			
Hsp27-F	TTCAGCCATGGCCGAGAGACGCATT	55	653
Hsp27-R	GTTGTAGTGCTCAGGGTTTCTTTG		
Primers for qRT-PCR			
Hsp27-F	CTCGGGAATGTCTGAGATAAAG	62	130
Hsp27-R	CTCATGTTTGCCGGTGAT		
Hsp60-F	GGAGAGCACAAACAGTGACTAC	62	130
Hsp60-R	GACACGGTCCTTCTTCTCATTC		
Hsp70-F	CTCTATGGTCCTGGTGAAGA	60	106
Hsp70-R	CCTCTGGGAGTCATTGAAATAG		
Hsp90-F	AGGTCACGGTCATCACTAAAC	62	182
Hsp90-R	GACCACTTCCTTCACTCTCTTC		
β-actin-F	GACCACCTTCAACTCCATCAT	62	126
β-actin-R	GTGATCTCCTTCTGCATCCTATC		

## Data Availability

Publicly available datasets were analyzed in this study. The rest of the data that are presented in this study are available on request from the corresponding author.

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
