# Peer review of "Schizothorax prenanti Heat Shock Protein 27 Gene: Cloning, Expression, and Comparison with Other Heat Shock Protein Genes after Poly (I:C) Induction"

_animals, 2022, doi:10.3390/ani12162034_

Round 1

Reviewer 1 Report

It was interesting to read the manuscript of Zhang et al. entitled Cloning of the Schizothorax prenanti heat shock protein 27 gene and expression levels of four S. prenanti heat shock protein genes after poly I:C challenge. Overall, this is piece of work that will interest the related research community and thus deserves a publication. However, I have some concerns and questions that should be addressed via revision before the article is formally published.

1. Extensive editing of English language and style were required.

2. Line 69-73, this part of descriptions confused me. Only the function of fish Hsp27 in response to viral challenge is unclear? How about the research progress on fish Hsp27, Hsp60, Hsp70, and Hsp90? The background on S. prenanti Hsps also needed to be clarified.

3. Line 141-159, at the qRT-PCR methodology section, did authors calculate the efficiency of primer pairs for the tested for SpHsps? please include the efficiency of each primer pairs used.

4. Line 167, 184, the headers should be modified, or reorganize the context of the results described, which should be corresponding strictly to the headers.

5. Line 361, please be careful to draw the conclusion on these four SpHsps may play important roles in the immune response and antiviral mechanism of S. prenanti. and avoid this tone in discussion section or any place. In fact, Hsps from mammals or fish were often hijacked to promote viral infection. It is not sufficient to predict its participation in the antiviral defense process only depending on the results of the gene expression level changes. 

Author Response

Comments and Suggestions for Authors:

Point 1: Extensive editing of English language and style were required.

Response 1: Dear Reviewer, thank you very much for your suggestion, the english language and style have been edited again. 

Point 2: Line 69-73, this part of descriptions confused me. Only the function of fish Hsp27 in response to viral challenge is unclear? How about the research progress on fish Hsp27, Hsp60, Hsp70, and Hsp90? The background on S. prenanti Hsps also needed to be clarified. 

Response 2: Dear Reviewer, we have adjusted the wording of this paragraph, actually, we mean, the function of S. prenanti Hsp27 in response to viral challenge is unclear. There are some reports about fish Hsp27, Hsp60, Hsp70, and Hsp90, These studies are compared in detail with the results of this study in Discussion of the MS, and, we added the background on S. prenanti Hsps in Introduction.

Point 3: Line 141-159, at the qRT-PCR methodology section, did authors calculate the efficiency of primer pairs for the tested for SpHsps? please include the efficiency of each primer pairs used.

Response 3: Dear Reviewer, we amplified each primer pairs and checked by agarose gel to confirm that the band was single and the size was in line with the size of the target fragment.

Point 4. Line 167, 184, the headers should be modified, or reorganize the context of the results described, which should be corresponding strictly to the headers.

Response 4: Dear Reviewer, thank you very much for your suggestion. We have modified the headers of Line 167, 184.

 Point 5. Line 361, please be careful to draw the conclusion on “these four SpHsps may play important roles in the immune response and antiviral mechanism of S. prenanti.” and avoid this tone in discussion section or any place. In fact, Hsps from mammals or fish were often hijacked to promote viral infection. It is not sufficient to predict its participation in the antiviral defense process only depending on the results of the gene expression level changes. 

Response 5: Dear Reviewer, we have revised the conclusion, included in the last sentence of the abstract. Which amended as these four SpHsps are involved in antiviral immunity and the response intensity of the four genes were organ-specific.

Reviewer 2 Report

The authors investigated the cloning of the Schizothorax prenanti heat shock protein 27 gene and expression levels of four S. prenanti heat shock protein genes after poly I:C challenge. This manuscript (MS) was clearly written and easy to understand. This work can help the sustainability of this species farming. However, some major issues significantly compromised the quality of this MS.

Major comments:

  • Title, you measured many hsps and is not clear why you only mentioned hsp27.
  • First, the manuscript needs to be edited by a native English speaker to improve the language of the MS and fix errors. I mentioned some of them, but still, more works needed to be done.
  • The authors should study much more about Hsps and some parts of the MS are not scientifically correct.
  • In the discussion, they should find and mention some studies where similar outputs were observed. The expression in muscle is higher than liver and is a bit strange for me.

However, I have touched on some more points that can contribute to the improvement of this MS.

Minor comments

Abstract

·       Please be consistent with the common name “Ya-fish” throughout the MS.

·       Line 16-19, please revise, further “breeding density” is not a common phrase.

·       Line 18-19, please revise “immune process”, does not make sense.

·       Line 21, first time in what and where? Animals, aquaculture?

·       Line 21-23, is not clear, please revise it. SpHsp27, SpHsp60, SpHsp70, and 22 SpHsp90 for the first time or only SpHsp27.

·       Line 25-26, how was different? Please avoid using this style of writing and be clear about how the treatments were changed, increased decreased unchanged.

·       Line 26-27, is not clear and  they are not the results of your study. In other words, is not relative to your work.

·       Line 35-37, please revise it.

·       “However, the response intensity of SpHsps was organ-specific” please review this part. Is not clear enough.

·       Line 47, is not scientifically correct. they have many duties that some of them are not relative to after stress.

·       Line 47-48, again is not scientifically correct. Please study more about these genes.

·       Line 58-62, please revise, you need to study more on this topic and many parts are not correct.

·       Line 71-72, 74, please be consistent with the common name. Further, here and throughout the MS, please first mention the common name plus the scientific name, and for the rest of the MS, just report the common name.

·       Line 141, “in unstressed conditions” better to say “normal condition”.

·       Line 163-164, please revise it.

·       Line 183, please see my comment regarding the common and scientific name.

·       Line 197-200, please revise it.

·       Figure 5, is quite strange for me. The liver should have much more expression than muscle. please clearly provide a reference that similarly, muscle had a higher level of hsps than liver.

·       In discussion section, there is no evidence that the trend of expression between tissues is in line with other studies.

·       Here and elsewhere, report P uppercase and italic (P<0.05).

·       Throughout the MS, if there is no significant difference, no need to report P-value.

·       Please reorder the keywords alphabetically and capitalize each word.

·       Please write the abstract more numerically about the results. You can do it by adding their numbers in parentheses.

·       Here and throughout the MS, please first mention the common name plus the scientific name, and for the rest of the MS, just report the common name.

·       Please update the introduction with recent works as many studies are available from the last two years, which were not included in this section.

·       Please review the literature much more carefully and cite more appropriate references.

·       Please mention the novelty of your work in the last paragraph of the introduction.

·       Some parts of the discussion are better updated with research in 2021 and 2022 as they refer to some old references. Please update the discussion with the latest studies as much as possible.

·       The conclusion needs to be revised and more comprehensive concepts  should be added there.

·        

Material and methods

·       Well-organized section. Clear fellow and all required details were provided.

·       Please mention how many percentages of water were exchanged each day if you have monitored.

·       For each analysis, please clarify how many fish were taken.

·       Please explain a little bit about your experimental diets, per each Table and Figure. Each Table and figure should represent enough information separately from the text.

·       Double-check the units and titles of all Tables and figures.

When revising your manuscript, please consider all issues mentioned in the reviewers' comments carefully with clear outlines for every change made in response to their comments including suitable rebuttals for any comments you deem inappropriate. Please itemize your response to each review comment, and highlight the revised at re-submission.

Best regards

Author Response

Major comments:

Title, you measured many hsps and is not clear why you only mentioned hsp27.

Response: Dear Reviewer, Another reviewer expressed a similar view, suggesting that the headline should read: ”Schizothorax prenanti heat shock protein 27 gene: cloning, expression, and comparison with other heat shock protein genes after Poly (I:C) induction”. We adopted the Title.

First, the manuscript needs to be edited by a native English speaker to improve the language of the MS and fix errors. I mentioned some of them, but still, more works needed to be done.

The authors should study much more about Hsps and some parts of the MS are not scientifically correct.

Response: Dear Reviewer, thank you very much for your suggestion. The MS was revised again by a native English speaker. At the same time, the Introduction and Discussion have been improved.

In the discussion, they should find and mention some studies where similar outputs were observed. The expression in muscle is higher than liver and is a bit strange for me.

However, I have touched on some more points that can contribute to the improvement of this MS.

Response: Dear Reviewer, We added relevant content in the Discussion. The expression in muscle is higher than liver, we reviewed the literature and concluded that the expression of Hsp27 gene in muscle and heart are higher than that in liver, as well as in kidney and spleen (shown in Discussion).

Minor comments:

Abstract

Please be consistent with the common name “Ya-fish” throughout the MS.

Response: Dear Reviewer, thank you very much for your suggestion. In fact, Ya-fish is the general name of Schizothorax prenanti and Schizothorax davidi, so we did not use Ya-fish throughout the MS. We also referred to other literatures, most of papers adopted S. prenanti, and some adopted Ya-fish, so, we think there can be no adjustment here, do you agree with us? Thank you!

Line 16-19, please revise, further “breeding density” is not a common phrase.

Response: Dear Reviewer, we have changed the “breeding density” to high-density culture.

Line 18-19, please revise “immune process”, does not make sense.

Response: Dear Reviewer, the “immune process” has been revised to immune mechanism

Line 21, first time in what and where? Animals, aquaculture?

Response: Dear Reviewer, I mean, first time in the fish Schizothorax prenanti. So "Hsp27" is preceded by the prefix "sp".

Line 21-23, is not clear, please revise it. SpHsp27, SpHsp60, SpHsp70, and SpHsp90 for the first time or only SpHsp27.

Response: Dear Reviewer, the gene SpHsp27 was cloned and characterised for the first time. Because SpHsp70 and SpHsp90 genes have been cloned and published, relevant literature of these two genes can be seen in the references of this MS.

Line 25-26, how was different? Please avoid using this style of writing and be clear about how the treatments were changed, increased decreased unchanged.

Response: Dear Reviewer, thank you very much for your suggestion. We changed “ different” to organ-specific. According to submission guidelines of the journal Animals, the summary is for amateurs, while abstract is for professionals. Therefore, the summary is too general to be described in detail due to the word limit. The detailed description is in the abstract and the article. 

Line 26-27, is not clear and they are not the results of your study. In other words, is not relative to your work.

Response: Dear Reviewer, we agree with you and have made adjustments: Our study is helpful to further understand the role of Hsp family genes in antiviral immunity mechanism of fish.

Line 35-37, please revise it.

“However, the response intensity of SpHsps was organ-specific” please review this part. Is not clear enough.

Response: Dear Reviewer, Line 35-39 is an interpretation of the response intensity of SpHsps was organ-specific. Of course, we have made some changes in the revised MS.

Line 47, is not scientifically correct. they have many duties that some of them are not relative to after stress.

Response: Dear Reviewer, we agree with you and some modifications have been done in the revised MS.

Line 47-48, again is not scientifically correct. Please study more about these genes.

Response: Dear Reviewer, we have revised the sentence in the MS.

Line 58-62, please revise, you need to study more on this topic and many parts are not correct.

Response: Dear Reviewer, we have revised the sentence.

Line 71-72, 74, please be consistent with the common name. Further, here and throughout the MS, please first mention the common name plus the scientific name, and for the rest of the MS, just report the common name.

Response: Dear Reviewer, we have revised the name.

Line 141, “in unstressed conditions” better to say “normal condition”.

Response: Thank you! We have modified it.

Line 163-164, please revise it.

Response: Dear Reviewer, we have revised it.

Line 183, please see my comment regarding the common and scientific name.

Response: Dear Reviewer, we have revised it.

Line 197-200, please revise it.

Response: Dear Reviewer, we have revised it.

Figure 5, is quite strange for me. The liver should have much more expression than muscle. please clearly provide a reference that similarly, muscle had a higher level of hsps than liver.

Response: Dear Reviewer, the expression in muscle is higher than liver, we reviewed the literature and concluded that the expression of Hsp27 gene in muscle and heart are higher than that in liver, as well as in kidney and spleen (shown in Discussion).

In discussion section, there is no evidence that the trend of expression between tissues is in line with other studies.

Response: Dear Reviewer, we have added relevant content in the discussion section.

Here and elsewhere, report P uppercase and italic (P<0.05).

Response: Dear Reviewer, we have checked it.

Throughout the MS, if there is no significant difference, no need to report P-value.

Response: Dear Reviewer, we have checked it.

Please reorder the keywords alphabetically and capitalize each word.

Response: Dear Reviewer, we have checked it.

Please write the abstract more numerically about the results. You can do it by adding their numbers in parentheses.

Response: Dear Reviewer, we have revised it.

Here and throughout the MS, please first mention the common name plus the scientific name, and for the rest of the MS, just report the common name.

Response: Dear Reviewer, we have revised it.

Please update the introduction with recent works as many studies are available from the last two years, which were not included in this section.

Response: Dear Reviewer, we have updated it.

Please review the literature much more carefully and cite more appropriate references.

Response: Dear Reviewer, thank you for your suggestion, and we have revised it.

Please mention the novelty of your work in the last paragraph of the introduction.

Response: thank you for your suggestion, yeah, we have have revised the last paragraph of the introduction.

Some parts of the discussion are better updated with research in 2021 and 2022 as they refer to some old references. Please update the discussion with the latest studies as much as possible.

Response: We have already updated part of the references, thank you for your suggestions.

The conclusion needs to be revised and more comprehensive concepts should be added there.

Response: Dear Reviewer, thank you for your suggestion, and we have revised the conclusion.

Material and methods

Well-organized section. Clear fellow and all required details were provided.

Please mention how many percentages of water were exchanged each day if you have monitored.

Response: Dear Reviewer, we have added it, about 1/4 of the water in each tank was exchanged daily.

For each analysis, please clarify how many fish were taken.

Response: Dear Reviewer, we have added the number of fish (n = 4) for each analysis.

Please explain a little bit about your experimental diets, per each Table and Figure. Each Table and figure should represent enough information separately from the text.

Response: Dear Reviewer, according to your suggestion, we have explained about the experimental diets in the MS, and some information was added to each Figure and Table.

Double-check the units and titles of all Tables and figures.

Response: Dear Reviewer, thank you four your suggestion, we have checked the Tables and Figures, and some changes have been done in the MS.

Reviewer 3 Report

Reviewers' Comments to Authors: 

The manuscript entitled “Cloning of the Schizothorax prenanti heat shock protein 27 gene and expression levels of four S. prenanti heat shock protein genes after poly I:C challenge” by Zhang et al. describes a scientific experiment to investigate molecular cloning and functional analysis of heat shock protein genes and their functional analysis under poly I:C treatment.

Based on scientific consideration, the manuscript contains some interesting findings that do contribute knowledge to the field of teleost fish. Generally, the manuscript is seeming to be acceptably written, however, when consider carefully it is found that the manuscript contains many major concerns and many trivial errors (such as grammatical errors, punctuation errors, etc.), and unclear points that the authors must address to warrant and improve the quality of the manuscript to reach acceptable standard levels for a scientific article. Additionally, to increase the better flow of the manuscript, the authors must pay attention to overhauling and presenting obtained data with better scientific patterns. The major and minor concerns are described below to improve the quality of the manuscript.

Title

To create better flow for this part and the whole manuscript, the title should be changed to such “Cloning of the Schizothorax prenanti heat shock protein 27 gene and its expression analysis compared with other three heat shock protein genes after poly I:C challenge”.

Simple summary

Line 17. Correct a grammatical error of “which” and the other place in the manuscript.

Line 19. Correct “Heat shock proteins” to “heat shock proteins” and keep consistent in using this form throughout.

Line 21. Correct “spHsp27” to “spHsp27” and keep consistent in using this form throughout. And “gene or protein” status forms must be rechecked thoroughly.

Line 26-27. The meaning and direction of the following sentence are unclear; “Our study is expected to contribute to disease prevention, aquatic medicine development, and the further development of S. prenanti culture.”.    

Abstract

Line 28 and 31. The authors tried to narrate that “gene” of SpHsp27 was cloned and characterize, however, in the manuscript just “cDNA” was analyzed. Therefore, this information must be revised in every part of the manuscript.

Line 40-41. Please consider the following carefully; “…these four SpHsps may play important roles in the immune response and antiviral mechanism of S. prenanti, especially the SpHsp27.”. The term “may play important roles” is too strong, it hard to believe that these molecules directly play their function on immune response or antiviral mechanism in fish!!

Introduction

Line 64. The logic of the following sentence is not true; “It is usually used to semulate immune activation during viral infections [12-13]”.

Line 70. Clarify the term “a virus”??

Line 72. Correct the scientific name of “Ietalurus punetaus” and the other species throughout.

The last paragraph of this section needs to be serially overhauled based on information presented in the manuscript. Additionally, information on severely viral diseases that cause harmful diseases in the target fish should be described to support the hypothesis and direction of the current research.

Methods and Methods

In this part, the authors should pay more attention to improving the quality of scientific writing, which was unsound.

2.1. Animal treatment

The authors should clearly describe an experimental design conducted in this part such as;

1) Number of fish in each group and water quality (other than temperature) must be declared. Did the authors maintain experimental fish individually?

2) Clearly identify the preparation of solution used to inject experimental fish. Was PBS used to dissolve poly I:C compared with the control that used PBS alone?? PBS pH should be also indicated.

3) It was unclear about the description of cDNA cloning of SpHSP27. If the authors want to clone the full cDNA of the target gene, why the author have to isolate total RNA from various organs, this is insane! 

4) Line 99. Correct “cm” to “cmˆ3”.

5) Line 106. Please carefully consider changing “quantitative real-time PCR (qPCR)” to quantitative real-time RT-PCR (qRT-PCR)” throughout the manuscript.  

6) Line 108. Please clarify the source of adipose tissues used in this part. Adipose tissues from various organs usually respond to their stimuli in different manners.   

2.2. RNA extraction and cDNA synthesis

In this part, did the authors use total RNA to synthesize 1st strand cDNA immediately or keep it in deep freeze temperature until needed?

2.3. Full-length cDNA cloning of the SpHsp27 gene

1) It was unclear why to authors had to recover the full cDNA sequence of the target gene, nevertheless full sequence was already discovered in transcriptome analysis. Please indicate the exact location of the used primers on the target cDNA. Additionally, it is not true to declare obtained sequence as “full length” if it was designed to cover from 5ÚTR to 3ÚTR. The term “partial cDNA” should be used instead of the authors having no information from 5ÚTR to poly A tail, please clarify.  

2) Why 1st a strand of cDNA from the liver used for this experiment??

3) Line 126. Correct “E.coli” to “Escherichia coli”, then italicize.  And how positive clones were selected??

“2.5. Tissue distribution of SpHsp27 mRNA in unstressed conditions” and “2.6. Detection of the expression patterns induced by poly I:C” should be merged and shortened by referring to the previous section properly. Additionally, the term “virus challenge” must be avoided using throughout, since “poly I:C” treatment is just a viral simulation.

2.7. Statistical analysis

1) Please serially revise the statistical analysis in this part step by step.

2) Please be consistent in using either “p or P-value” throughout the manuscript.   

Results

1) Based on the previous comments and results observed in Figure 1. It was clear that the information of SpHsp27 is just partial cDNA, the authors must be vigorously revised this information in every part of the manuscript. Additionally, important motifs or any other crucial domains of SpHsp27 molecules must be characterized to increase and impact and quality of the manuscript.  

2) In Figure 1, nucleotide and amino acid sequences should be numbered!

3) Please revise the description of Figure 2a and correct Helix, Coil, and Strand by uncapitalized changes.

4) Figure 3. Description of data obtained in Figure 3 is too brief to understand.

5) 3.2 Phylogenetic analysis.

- In this part, information on homological analysis should be moved to be placed in 3.1. Additionally, the authors previously declared in “Materials and Methods” that the tree was generated by cDNA sequences, but amino acid sequences in this part???

- Why C. carpio was not seen in the tree?

- The proper out group should be placed in the evolutionary tree!!

6) Figure 4. Put ”after species names” at the end of its legends.

7) 3.3.Tissue distribution of SpHsp27 expression in S. prenanti.

Line 197. Please correct the following sentence; “β-actin was used as the internal control”, since β-actin was just not used as the internal control, but for normalization to provide a relative expression ratio as well.

8) Figure 5. Please carefully recheck the statistical analysis of this figure and in the figure legend, please correct “a, b” to “a, b and c” or any others proper.

9) Figures 6, 7, 8, and 9. Please clearly indicate a calibrator used in “Materials and Methods” to calculate fold change or relative expression ratio in these figures.  

10) Line 249. Please revise the following sentence; “….up-regulated at 12 h and 24 h after poly I:C treatment.”, since it is the wrong description.

11) 272-275. Please reconsider removing the following information to “Discussion” or delete it from the manuscript; “The spleen is one of the main immune organs, but no significant changes in the expression level of SpHsp90 were found within 24 h of poly I:C injection. We speculate that SpHsp90 gene expression in the spleen was involved in the immune response at a later time. However, this needs be confirmed by further investigation.”.

Discussion

1) In this part, a comparison analysis of all of the characteristics of the “SpHsp27” must be discussed properly. The evolutionary study must be also narrated. And please mainly target the SpHsp27 than other Hsp molecules.

2) Line 289-292. Discussion in this part is informative, since information of Stenberg et al. (2019) used “head kidney leukocytes (HKLs)” as the target tissues, while the current study used whole kidney tissues to investigate whether that is more complex than that of HKLs.

3) The following sentence is very awkward; “Therefore, we conclude that Hsp60 plays a similar role in S. prenanti.”.

4) Line 331. Italicize “Streptococcus agalactiae”.

5) Please carefully revise the following information; “In summary, the SpHsp90 gene may be involved in resistance to both bacterial and viral challenges and may be an early marker of infection.”. Since no bacterial and viral infection experiments were conducted in the current study.

6) Line The following information is exaggerated; “….most suitable organs for examining SpHsp27 expression,…”.

Conclusion

Line 359. Please correct the grammatical error of “which was”.

Institutional Review Board Statement: The experimental fish S. prenanti used in this study were 369 obtained from a commercial aqua-farm and therefore did not require ethical approval.

Please correct the logic of this information.

References

There are some formational errors observed in this part, such as the abbreviation of journal format, scientific name, and common names. Such as references 8, 15, 17, 19, 22, 27, 34, 42.

Author Response

Title

To create better flow for this part and the whole manuscript, the title should be changed to such “Cloning of the Schizothorax prenanti heat shock protein 27 gene and its expression analysis compared with other three heat shock protein genes after poly I:C challenge”.

Response: Dear Reviewer, thank you four your suggestion and, we agree with you. Taking into consideration the opinions of other reviewers and language editors, we will revise the title as: Schizothorax prenanti heat shock protein 27 gene: cloning, expression, and comparison with other heat shock protein genes after Poly (I:C) induction.

Simple summary

Line 17. Correct a grammatical error of “which” and the other place in the manuscript.

Response: Dear Reviewer, thank you four your suggestion, we have revised the sentences. 

Line 19. Correct “Heat shock proteins” to “heat shock proteins” and keep consistent in using this form throughout.

Response: Dear Reviewer, we have revised it.

Line 21. Correct “spHsp27” to “spHsp27” and keep consistent in using this form throughout. And “gene or protein” status forms must be rechecked thoroughly.

Response: Dear Reviewer, we have revised it.

Line 26-27. The meaning and direction of the following sentence are unclear; “Our study is expected to contribute to disease prevention, aquatic medicine development, and the further development of S. prenanti culture.”.    

Response: Dear Reviewer, another reviewer also listed similar question, and we have revised the sentences, thank you!

Abstract

Line 28 and 31. The authors tried to narrate that “gene” of SpHsp27 was cloned and characterize, however, in the manuscript just “cDNA” was analyzed. Therefore, this information must be revised in every part of the manuscript.

Response: Dear Reviewer, we have corrected it in the MS.

Line 40-41. Please consider the following carefully; “…these four SpHsps may play important roles in the immune response and antiviral mechanism of S. prenanti, especially the SpHsp27.”. The term “may play important roles” is too strong, it hard to believe that these molecules directly play their function on immune response or antiviral mechanism in fish!!

Response: Dear Reviewer, another reviewer also listed the similar question, and we have revised the sentences in the MS, thank you!

Introduction

Line 64. The logic of the following sentence is not true; “It is usually used to semulate immune activation during viral infections [12-13]”.

Response: Dear Reviewer, the sentence has been changed to It is usually used as a viral analog to infect organisms

Line 70. Clarify the term “a virus”??

Response: Dear Reviewer, the sentence has been changed to To investigate the temporal expression levels of Hsp genes in S. prenanti after virus infection

Line 72. Correct the scientific name of “Ietalurus punetaus” and the other species throughout.

Response: Dear Reviewer, we have revised“Ietalurus punetaus”to Ictalurus punctatus.

The last paragraph of this section needs to be serially overhauled based on information presented in the manuscript. Additionally, information on severely viral diseases that cause harmful diseases in the target fish should be described to support the hypothesis and direction of the current research.

Response: Dear Reviewer, we have revised the last paragraph of the introduction.

Methods and Methods

In this part, the authors should pay more attention to improving the quality of scientific writing, which was unsound.

Response: Dear Reviewer, thank you for you suggestion, some corresponding modifications have been made.

2.1. Animal treatment

The authors should clearly describe an experimental design conducted in this part such as;

1) Number of fish in each group and water quality (other than temperature) must be declared. Did the authors maintain experimental fish individually?

Response: Dear Reviewer, the number of fish in each group was 4, as indicated in Figures 5 to 9, (n = 4 fish per group). We have added some new information in section of Animal treatment.

As for water quality, unfortunately, we only monitored temperature, we will pay attention to monitoring more water quality in future studies.

As for the experimental fish, we cultured them in a glass tank. After the 10 days acclimation, 4 fish in the control group were injected with PBS and then anesthetized and dissected, while the other 8 fish were injected with Poly I:C and anesthetized and dissected 12 and 24 h post-Poly I:C injection, respectively, seen in 2.1. Animal treatment of the MS.

2) Clearly identify the preparation of solution used to inject experimental fish. Was PBS used to dissolve poly I:C compared with the control that used PBS alone?? PBS pH should be also indicated.

Response: Dear Reviewer, both the PBS (HyClone) and poly I:C (P1530, sigma, St Louis, USA) were purchased from reagent Companies, and used separately. seen in 2.1. Animal treatment of the MS.

3) It was unclear about the description of cDNA cloning of SpHSP27. If the authors want to clone the full cDNA of the target gene, why the author have to isolate total RNA from various organs, this is insane!

Response: Dear Reviewer, we used liver as a template to clone the complete coding sequence of the SpHSP27. And we isolated total RNA from various organs to detect the different tissue expression of SpHSP27.

4) Line 99. Correct “cm” to “cmˆ3”.

Response: Dear Reviewer, we have corrected cm to cm3.

5) Line 106. Please carefully consider changing “quantitative real-time PCR (qPCR)” to “quantitative real-time RT-PCR (qRT-PCR)” throughout the manuscript.  Response: Dear Reviewer, thank you four your suggestion, we have revised them.

  • Line 108. Please clarify the source of adipose tissues used in this part. Adipose tissues from various organs usually respond to their stimuli in different manners.
  • Response: Dear Reviewer, we have changed the ‘adipose’to ‘intraperitoneal fat’ in the whole MS.

2.2. RNA extraction and cDNA synthesis

In this part, did the authors use total RNA to synthesize 1st strand cDNA immediately or keep it in deep freeze temperature until needed?

Response: Dear Reviewer, we used total RNA to synthesize 1st strand cDNA immediately and the remaining total RNA were kept in deep freeze temperature until needed.

2.3. Full-length cDNA cloning of the SpHsp27 gene

1) It was unclear why to authors had to recover the full cDNA sequence of the target gene, nevertheless full sequence was already discovered in transcriptome analysis. Please indicate the exact location of the used primers on the target cDNA. Additionally, it is not true to declare obtained sequence as “full length” if it was designed to cover from 5ÚTR to 3ÚTR. The term “partial cDNA” should be used instead of the authors having no information from 5ÚTR to poly A tail, please clarify.  

Response: Dear Reviewer, we have replaced the full-length with partial cDNA. We cloned the complete coding sequence of spHsp27 according to transcriptome data. On the one hand, we confirmed the accuracy of spHsp27 in the transcriptome, and on the other hand, obtaining the complete coding sequence can more accurately design quantitative primers, ensuring that the primers are in coding region.

2) Why 1st a strand of cDNA from the liver used for this experiment??

Response: Dear Reviewer, although the expression abundance of hsp27 in liver is not the highest, but the relative content of RNA in liver is high, we can obtain a sufficient amount of total RNA for gene cloning. The collected tissues collected are used for other detections, except real-time quantitative detection. In order to meet the requirements of all experimental tissue volume, 1st a strand of cDNA from the liver used for this experiment.

3) Line 126. Correct “E.coli” to “Escherichia coli”, then italicize. And how positive clones were selected??

Response: Dear Reviewer, we have corrected“E.coli”to“Escherichia coli”.

We used ampicillin to screen positive clones, after that, we selected positive bacteria for PCR amplification, and then sent them for sequencing to confirm whether they were correct.

 “2.5. Tissue distribution of SpHsp27 mRNA in unstressed conditions” and “2.6. Detection of the expression patterns induced by poly I:C” should be merged and shortened by referring to the previous section properly. Additionally, the term “virus challenge” must be avoided using throughout, since “poly I:C” treatment is just a viral simulation.

Response: Dear Reviewer, we have corrected“virus challenge”to“poly I:C challenge”.

2.5 and 2.6are two relatively independent contents, the title will be long after merging, we do not suggest merge, do you agree with us?

2.7. Statistical analysis

1) Please serially revise the statistical analysis in this part step by step.

  • Please be consistent in using either “p or P-value” throughout the manuscript.

Response: Dear Reviewer, we use P-value throughout the and corrected the p.

Results

1) Based on the previous comments and results observed in Figure 1. It was clear that the information of SpHsp27 is just partial cDNA, the authors must be vigorously revised this information in every part of the manuscript. Additionally, important motifs or any other crucial domains of SpHsp27 molecules must be characterized to increase and impact and quality of the manuscript.  

Response: Dear Reviewer, we have replaced the full-length with partial cDNA. And we have added crucial domains of SpHsp27 molecules in MS.

2) In Figure 1, nucleotide and amino acid sequences should be numbered!

Response: Dear Reviewer, the Figure 1 has been replaced.

3) Please revise the description of Figure 2a and correct Helix, Coil, and Strand by uncapitalized changes.

Response: Dear Reviewer, we have corrected Helix, Coil, and Strand by uncapitalized changes.

4) Figure 3. Description of data obtained in Figure 3 is too brief to understand.

Response: Dear Reviewer, thank you for you suggestion. We have added analysis to Figure 3 in MS.

5) 3.2 Phylogenetic analysis.

- In this part, information on homological analysis should be moved to be placed in 3.1. Additionally, the authors previously declared in “Materials and Methods” that the tree was generated by cDNA sequences, but amino acid sequences in this part???

Response: Dear Reviewer, we have moved homological analysis to be placed in 3.1. About “Materials and Methods”that the tree was generated by cDNA sequences, but amino acid sequences in this part, this is a mistake we made. We have replaced the cDNA sequences with amino acid sequences.  

- Why C. carpio was not seen in the tree?

Response: Thank you for your comment, this is our mistake. We have replaced C.carpio with Carassius auratus. Secondly, NCBI database did not have the full sequence of Hsp27 of C. carpio, but only the full sequence of a suspected common carp (marked -like). For conservative reasons, we did not select the suspected sequence when making phylogenetic tree

- The proper out group should be placed in the evolutionary tree!!

Response: Dear Reviewer, thank you for your suggestion. We have added the proper out group  in the evolutionary tree.

6) Figure 4. Put “after species names” at the end of its legends.

Response: Dear Reviewer, we have corrected it in MS.

7) 3.3.Tissue distribution of SpHsp27 expression in S. prenanti.

Line 197. Please correct the following sentence; “β-actin was used as the internal control”, since β-actin was just not used as the internal control, but for normalization to provide a relative expression ratio as well.

ResponseDear Reviewer, we have corrected it in the MS.

8) Figure 5. Please carefully recheck the statistical analysis of this figure and in the figure legend, please correct “a, b” to “a, b and c” or any others proper.

Response: Dear Reviewer, thank you for your suggestion, we have corrected it in the MS.

9) Figures 6, 7, 8, and 9. Please clearly indicate a calibrator used in “Materials and Methods” to calculate fold change or relative expression ratio in these figures.  

Response:  Dear Reviewer, we have added it in MS.

10) Line 249. Please revise the following sentence; “….up-regulated at 12 h and 24 h after poly I:C treatment.”, since it is the wrong description.

Response: Dear Reviewer, thank you for your advice, we have revised the sentence.

11) 272-275. Please reconsider removing the following information to “Discussion” or delete it from the manuscript; “The spleen is one of the main immune organs, but no significant changes in the expression level of SpHsp90 were found within 24 h of poly I:C injection. We speculate that SpHsp90 gene expression in the spleen was involved in the immune response at a later time. However, this needs be confirmed by further investigation.”.

Response: Dear Reviewer, thank you for your advice, we removed the sentence to “Discussion”.

Discussion

1) In this part, a comparison analysis of all of the characteristics of the “SpHsp27” must be discussed properly. The evolutionary study must be also narrated. And please mainly target the SpHsp27 than other Hsp molecules.

Response: Dear Reviewer, thank you for your advice,. We have added relative discussion.

 2) Line 289-292. Discussion in this part is informative, since information of Stenberg et al. (2019) used “head kidney leukocytes (HKLs)” as the target tissues, while the current study used whole kidney tissues to investigate whether that is more complex than that of HKLs.

3) The following sentence is very awkward; “Therefore, we conclude that Hsp60 plays a similar role in S. prenanti.”.

Response: Dear Reviewer, thank you for your advice, we revised the sentence to Therefore, we infer that Hsp60 plays a similar but organ-specific role in the anti-pathogen immunity of S. prenanti

4) Line 331. Italicize “Streptococcus agalactiae”.

Response: thank you for your advice.

5) Please carefully revise the following information; “In summary, the SpHsp90 gene may be involved in resistance to both bacterial and viral challenges and may be an early marker of infection.”. Since no bacterial and viral infection experiments were conducted in the current study.

Response: Dear Reviewer, In combination with the above sentences, we combined the study of other researcher, Pu et al. (2016) reported that SpHsp90 was involved in the immunity of S. prenanti to bacteria (Streptococcus agalactiae), and the present study found that SpHsp90 was activated by virus analog (poly I:C ).

6) Line The following information is exaggerated; “….most suitable organs for examining SpHsp27 expression,…”.

Response: Dear Reviewer, we agree with your opinion, and some changes have been made.

Conclusion

Line 359. Please correct the grammatical error of “which was”.

Response: Dear Reviewer, we have correct the grammatical error.

Institutional Review Board Statement: The experimental fish S. prenanti used in this study were obtained from a commercial aqua-farm and therefore did not require ethical approval.

Please correct the logic of this information.

Response: Dear Reviewer, we have revised the Institutional Review Board Statement in the MS.

References

There are some formational errors observed in this part, such as the abbreviation of journal format, scientific name, and common names. Such as references 8, 15, 17, 19, 22, 27, 34, 42.

Response: Dear Reviewer, we have revised the references, thank you for your advice.

Round 2

Reviewer 1 Report

The authors have fixed issues of presentation and, with a few exceptions that can be corrected, are no longer making unjustified claims. However, my fundamental critique remains that the manuscript lacks sufficiently substantive and reliable data to support the conclusion that SpHsps are involved in the antiviral immunity. Please avoid this tone in discussion section or any place in the manuscript. In fact, Hsps were often hijacked to promote viral infection. It is not sufficient to predict its participation in the antiviral defense process only depending on the results of the gene expression level changes. They may mediate pathological process if they are hijacked by virus and induced with high expression levels. This is not antiviral.

  •  

Author Response

Dear Reviewer, thank you very much for your careful review of our manuscript. We all authors agree with your suggestion, and, we have avoid the positive tone in the manuscript and describe it as speculative tone, such as “Our findings indicated that Poly (I:C) induced SpHsp27, SpHsp60, SpHsp70, and SpHsp90 expression and these organ-specific SpHsps are potentially involved in S. prenanti antiviral immunity or mediate pathological process.”

Reviewer 2 Report

The authors have not improved the quality of the MS and I suggest authors reading one more time comments carefully and check all details. Further, please clarify in which line you addressed comments for any single comment from reviewers. The reviewer can not see how you improved the comment. For example, in my comment regarding liver and muscle, you said it is in discussion but I read this section and I could not find it.

Author Response

Comments and Suggestions for Authors:

The authors have not improved the quality of the MS and I suggest authors reading one more time comments carefully and check all details. Further, please clarify in which line you addressed comments for any single comment from reviewers. The reviewer can not see how you improved the comment. For example, in my comment regarding liver and muscle, you said it is in discussion but I read this section and I could not find it.

Response 1: Dear Reviewer, we are very sorry that the reply letter of Round 1 did not indicate completely which line replied your comments and suggestions.

Actually, in the Round 1 of Comments and Suggestions for Authors, the line number of your MS version has some deviation from our original manuscript, that is, the line number you proposed to modify is not corresponding to our manuscript. For example, "Line 141," in unstressed conditions "better to say" normal condition". in fact, we find "normal condition" in Line 143, the line number is off by two lines, but elsewhere it's off by more than two lines. For another example, you mentioned "Line 197-200, please revise it." But we could not find the corresponding problem in Line 197-200 of the MS. We speculate that it may mean "2.7. Statistical Analysis". There are many such problems, which trouble us authors.

There are three reviewers, and the line numbers of the other two are the same as those of our original manuscript. We speculate that the document software you use may be different from ours. however, if you use the PDF version of the MS, the line numbers will be the same. 

Now we will provide additional information on some replies that were not specify a specific line number in the Round 1.

Comments and Suggestions for Authors in Round 1:

Major comments:

In the discussion, they should find and mention some studies where similar outputs were observed. The expression in muscle is higher than liver and is a bit strange for me.

However, I have touched on some more points that can contribute to the improvement of this MS.

Response:

Dear Reviewer, we are so sorry that we did not complete the discussion in the Round 1 revision. Although we have attached relevant references 45-48 in Round 1 , but we forgot to add this content in Discussion. Now we have added it, sorry again. The expression in muscle is higher than liver, we reviewed the references and concluded that the expression of Hsp27 gene in muscle and heart are higher than that in liver, as well as in kidney and spleen (shown in Discussion, Line 313-319).

Minor comments:

Abstract

Line 16-19, please revise, further “breeding density” is not a common phrase.

Response: Dear Reviewer, we have changed the “breeding density” to “high-density farming”. Line 17, Line 96.

Line 18-19, please revise “immune process”, does not make sense.

Response: Dear Reviewer, the “immune process” has been revised to “immune mechanism”. Line 19.

Line 21-23, is not clear, please revise it. SpHsp27, SpHsp60, SpHsp70, and SpHsp90 for the first time or only SpHsp27.

Response: Dear Reviewer, the gene SpHsp27 was cloned and characterised for the first time. Because SpHsp70 and SpHsp90 genes have been cloned and published, relevant literature of these two genes can be seen in the references of this MS. Line 455, Line 458

Line 25-26, how was different? Please avoid using this style of writing and be clear about how the treatments were changed, increased decreased unchanged.

Response: Dear Reviewer, thank you very much for your suggestion. We have changed “ different” to “organ-specific”. Line 27.

Line 26-27, is not clear and they are not the results of your study. In other words, is not relative to your work.

Response: Dear Reviewer, we agree with you and have made adjustments: Our study is helpful to further understand the role of Hsp family genes in antiviral immunity mechanism of fish. Line28-29.

Line 35-37, please revise it.

“However, the response intensity of SpHsps was organ-specific” please review this part. Is not clear enough.

Response: Dear Reviewer, Line 39-42 is an interpretation of the “response intensity of SpHsps was organ-specific”. Of course, we have made some changes in the revised MS. Line 39-42.

Line 47, is not scientifically correct. they have many duties that some of them are not relative to after stress.

Response: Dear Reviewer, we agree with you and added a limiting word "generally" in the revised MS. Line 51.

Line 47-48, again is not scientifically correct. Please study more about these genes.

Response: Dear Reviewer, we have revised the sentence in the MS. Line 52-53.

Line 58-62, please revise, you need to study more on this topic and many parts are not correct.

Response: Dear Reviewer, we have revised the sentence. Line 63-67.

Line 71-72, 74, please be consistent with the common name. Further, here and throughout the MS, please first mention the common name plus the scientific name, and for the rest of the MS, just report the common name.

Response: Dear Reviewer, we have revised the name. Line 81-82.

Line 141, “in unstressed conditions” better to say “normal condition”.

Response: Thank you! We have modified it. Line 156.

Line 163-164, please revise it.

Response: Dear Reviewer, we have revised it. Line 180-183.

Line 183, please see my comment regarding the common and scientific name.

Response: Dear Reviewer, we have revised it. Line 192.

Line 197-200, please revise it.

Response: Dear Reviewer, we have revised it. Line 223-226.

Figure 5, is quite strange for me. The liver should have much more expression than muscle. please clearly provide a reference that similarly, muscle had a higher level of hsps than liver.

Response: Dear Reviewer, the expression in muscle is higher than liver, we reviewed the literature and concluded that the expression of Hsp27 gene in muscle and heart are higher than that in liver, as well as in kidney and spleen (shown in Discussion, Line 313-319).

In discussion section, there is no evidence that the trend of expression between tissues is in line with other studies.

Response: Dear Reviewer, we have added relevant content in the discussion section, and there are evidences that the trend of expression between tissues is in line with other studies. Line 313-319

Please reorder the keywords alphabetically and capitalize each word.

Response: Dear Reviewer, we have reordered the keywords alphabetically, Line 46, however, each word of the keywords do not need to be capitalized according to Animals Microsoft Word template.

Please write the abstract more numerically about the results. You can do it by adding their numbers in parentheses. 

Response: Dear Reviewer, we have revised it. Line 39-43.

Please update the introduction with recent works as many studies are available from the last two years, which were not included in this section.

Response: Dear Reviewer, we have updated some of the references, which are the researches of the last two years, such as references 2, 6, 10, 12, 31.

Please mention the novelty of your work in the last paragraph of the introduction.

Response: thank you for your suggestion, yeah, we have have revised the last paragraph of the introduction. Line 100-103.

Some parts of the discussion are better updated with research in 2021 and 2022 as they refer to some old references. Please update the discussion with the latest studies as much as possible.

Response: Thank you for your suggestion, we have already updated part of the references, such as references 44, 47, 49, 60. It should be noted that although some references we cited are older, they are authoritative in the field of HSP-related studies, so we cite these references. Of course, we have also searched studies on HSP in the past two years, but they are not closely related to the Discussed in our study.

The conclusion needs to be revised and more comprehensive concepts should be added there.

Response: Dear Reviewer, thank you for your suggestion, and we have revised the conclusion. Line388-395.

Additionally, our manuscript has been revised by professional companies for two rounds of grammar revision, and the "Author 1" in the revised manuscript is a sign of the second revision in grammar. Now, we have check and modify the grammar again in Round 2.

Reviewer 3 Report

Reviewers' Comments to Authors:

The manuscript entitled “Schizothorax prenanti heat shock protein 27 gene: cloning, expression, and comparison with other heat shock protein genes after Poly (I:C) induction” by Zhang et al. describes a scientific experiment to investigate molecular cloning and functional analysis of heat shock protein genes and their functional analysis under poly I:C treatment.

The revised manuscript has almost responded all comments raised in previous suggestions. However, it still contains some errors that the authors should properly address. The recommendation of “accept” should be suggested after the following comments would be improved.

Simple summary

Line 19. Correct a grammatical error of “which” and the other place in the manuscript.

Line 21-22. Revise the following sentence “Here, we cloned and identified SpHsp27 using polyinosinic-polycytidylic acid (Poly (I:C)) as a viral analog”. How “polyinosinic-polycytidylic acid (Poly (I:C)) could be used as a tool to clone and identified the SpHsp27”???

(Poly (I:C)) should be replaced with “[Poly (I:C)]” throughout.

Additionally, the term “cDNA encoding SpHsp27 genes” should be considered to add in this content.

Abstract

Line 28. The following content; “We identified and cloned the heat shock protein (Hsp) 27 gene…” should be replaced with “We identified and cloned cDNA encoding the heat shock protein (Hsp) 27 gene…”

Line 32. Delete “a gene”.

Keywords

Correct “Heat shock protein 27.” to “Heat shock protein 27,”.  

Introduction

Line 74. “…, it is in aquaculture” unclear and needs reference(s).

Line 78. Correct the genus name of “Ietalurus” and the other species throughout.

Line 96. Correct “…and cloned SpHsp27..” to “…and cloned cDNA encoding SpHsp27 gene...”   

Methods and Methods

2.7. Statistical analysis

Please serially revise statistical analysis in this part step by step. The information that the authors descripted is not acceptable!!

Results

1) Figure 1, amino acid sequence should be numbered as [XX]!

Conclusions

The following content should be considered to add at the beginning of this section; “In the present study, the partial cDNA encoding the full ORF of SpHsp27 genes of S. prenanti was successfully cloned and characterized.”.

References

There are some formational errors observed in this part, such as the abbreviation of journal format, scientific name, and common names. Such as references 12, 31, 35, 44.

Author Response

Simple summary

Point 1: Line 19. Correct a grammatical error of “which” and the other place in the manuscript.

Response 1: Dear Reviewer, we have changed “which results in economic losses” to “resulted in economic losses”. Line 19, and deleted “which is” in Line 107.

Point 2: Line 21-22. Revise the following sentence “Here, we cloned and identified SpHsp27 using polyinosinic-polycytidylic acid (Poly (I:C)) as a viral analog”. How “polyinosinic-polycytidylic acid (Poly (I:C)) could be used as a tool to clone and identified the SpHsp27”???

Response 2: Dear Reviewer, we have revised the sentence: “we identified and cloned cDNA encoding SpHsp27 gene and detected its tissue distribution, and using polyinosinic-polycytidylic acid [Poly (I:C)] as a viral analog to challenge the fish.

Point 3: (Poly (I:C)) should be replaced with “[Poly (I:C)]” throughout.

Response 3: Dear Reviewer, we have replaced it.

Point 4: Additionally, the term “cDNA encoding SpHsp27 genes” should be considered to add in this content.

Response 4: Dear Reviewer, we have added the term.

Abstract

Point 5: Line 28. The following content; “We identified and cloned the heat shock protein (Hsp) 27 gene…” should be replaced with “We identified and cloned cDNA encoding the heat shock protein (Hsp) 27 gene…”

Response 5: Dear Reviewer, we agree with you, thank you for your suggestion.

Point 6: Line 32. Delete “a gene”.

Response 6: Dear Reviewer, we have deleted it, thank you for your suggestion.

Point 7: Keywords  Correct “Heat shock protein 27.” to “Heat shock protein 27,”.

Response 7: Dear Reviewer, we have corrected the Keywords, thank you for your suggestion.

Introduction

Point 8: Line 74. “…, it is in aquaculture” unclear and needs reference(s).

Response 8: Dear Reviewer, we have revised the sentence “it has been used to study the antiviral mechanism of aquatic animals [23-36]. The role of Poly (I:C) in enhancing natural immunity and reducing disease risk has attracted attention.

Point 9: Line 78. Correct the genus name of “Ietalurus” and the other species throughout.

Response 9: Dear Reviewer, we have corrected Ietalurus to Ictalurus, thank you for your suggestion.

Point 10: Line 96. Correct “…and cloned SpHsp27..” to “…and cloned cDNA encoding SpHsp27 gene...”

Response 10: Dear Reviewer, we have corrected the sentence, thank you for your suggestion.

Methods and Methods

2.7. Statistical analysis

Point 11: Please serially revise statistical analysis in this part step by step. The information that the authors descripted is not acceptable!!

Response 11: Dear Reviewer, thank you for your suggestion. In fact, we referred to the statistical method of Du Xiaogang et al., [Multiple subtypes of TLR22 molecule from Schizothorax prenanti present the functional diversity in ligand recognition and signal activation, Fish and Shellfish Immunology, 2019, 93, 986-996, https://doi.org/10.1016/j.fsi.2019.08.042]. 

And , we have revise the statistical analysis as: SPSS 22.0 and Graphpad Prism 5.0 software was used for date analysis and histogram, respectively. The mRNA expression levels were analyzed by using One-way ANOVA method. All data are presented as the mean ± standard error (n = 4), and the statistically significant differences between PBS control and Poly (I:C) treatment groups at each time point are expressed with asterisks,* P < 0.05, ** P < 0.01, *** P < 0.001 vs. corresponding control group at the time points.

Point 12: Results

  • Figure 1, amino acid sequence should be numbered as [XX]!

Response 12: Dear Reviewer, the amino acid sequence have be numbered as [XX], thank you for your suggestion.

Point 13: Conclusions

  The following content should be considered to add at the beginning of this section; “In the present study, the partial cDNA encoding the full ORF of SpHsp27 genes of S. prenanti was successfully cloned and characterized.”.

Response 13: Dear Reviewer, we agree with you, thank you for your suggestion.

References

Point 14: There are some formational errors observed in this part, such as the abbreviation of journal format, scientific name, and common names. Such as references 12, 31, 35, 44.

Response 14: Dear Reviewer, We have carefully checked all the references again and corrected those errors, thank you for your suggestion.

Round 3

Reviewer 1 Report

The authors have responded to all comments and I recommend the publication of the revised manuscript.

Reviewer 2 Report

The authors improved the quality of MS and now can go to further process.